# Comparative Transcriptomics Reveals Novel and Differential Circular RNA Responses Underlying Interferon-Mediated Antiviral Regulation in Porcine Alveolar Macrophages

**DOI:** 10.3390/v17101307

**Published:** 2025-09-27

**Authors:** Jiuyi Li, Oluwaseun Adeyemi, Laura C. Miller, Yongming Sang

**Affiliations:** 1Department of Food and Animal Sciences, College of Agriculture, Tennessee State University, Nashville, TN 37209, USA; jli4@tnstate.edu (J.L.); oadeyemi@my.tnstate.edu (O.A.); 2Diagnostic Medicine and Pathobiology, College of Veterinary Medicine, Kansas State University, 1800 Denison Ave, Manhattan, KS 66506, USA; lauramiller@ksu.edu

**Keywords:** Porcine Reproductive and Respiratory Syndrome, PRRS virus, alveolar macrophage, circular RNA, interferon

## Abstract

Porcine Reproductive and Respiratory Syndrome (PRRS) causes significant economic losses in the swine industry. Circular RNAs (circRNAs), a class of stable non-coding RNAs, are increasingly recognized as regulators in immune responses and host–virus interactions. This study investigated the genome-wide circRNA responses in porcine alveolar macrophages (PAMs), key cell targets of PRRSV, following treatment with a modified live virus (MLV) vaccine or two interferon (IFN) subtypes (IFN-α1, IFN-ω5). Using RNA sequencing, we identified over 1000 differentially expressed circRNAs across treatment groups, revealing both conserved and distinct expression profiles. Gene Ontology and KEGG pathway analyses indicated that circRNA-associated genes are significantly enriched in immune-related processes and pathways, including cytokine signaling and antiviral defense. Notably, IFN-ω5 treatment induced a pronounced circRNA response, aligning with its potent antiviral activity. We further explored the regulatory potential of these circRNAs by predicting miRNA binding sites, revealing complex circRNA-miRNA interaction networks. Additionally, we assessed the coding potential of differentially expressed circRNAs by identifying open reading frames (ORFs), internal ribosome entry sites (IRESs), and N6-methyladenosine (m^6^A) modification sites, suggesting a subset may undergo non-canonical translation. These findings provide a comprehensive landscape of circRNA expression in PAMs under different antiviral conditions, highlighting their potential roles as immune regulators and novel players in interferon-mediated antiviral responses, particularly downstream of IFN-ω5. This work contributes to understanding the non-coding RNA landscape in the PRRSV-swine model and suggests circRNAs as potential targets for future antiviral strategies.

## 1. Introduction

Porcine Reproductive and Respiratory Syndrome (PRRS) is a highly contagious viral disease that causes significant economic losses in the swine industry by inducing respiratory distress in piglets and reproductive failure in sows [1]. The economic impact of PRRS is well-documented, with direct annual losses in the United States alone estimated at over USD 600 million [2]. The causative agent, PRRS virus (PRRSV), is an enveloped, single-stranded RNA virus in the *Arteriviridae* family, sharing some genomic structural features with coronaviruses [3]. Two genetically distinct species, Type 1 (European) and Type 2 (North American), have emerged and spread globally [4]. The PRRSV genome is approximately 15 kb in length. About 75% of the genome, located at the 5′ terminus, comprises open reading frames *ORF1a* and *ORF1b*, which encode the viral polyproteins pp1a and pp1ab [5]. These polyproteins are subsequently cleaved into non-structural proteins (Nsps) involved in viral replication and host pathogenesis. The remaining 25% of the genome, located at the 3′ terminus, contains eight ORFs that encode structural proteins, including the envelope (E) protein, glycoproteins GP2–GP5, membrane (M) protein, and nucleocapsid (N) protein [6].

During acute PRRSV infection, the virus primarily targets porcine monocytic cells including blood monocytes, monocyte-derived dendritic cells (mDCs), and particularly in PAMs, leading to their disruption [7]. As sentinel immune cells residing in the alveolar space, PAMs are key players in initiating immune responses. Their surfaces express molecules that modulate phagocytic function and efferocytosis—the clearance of apoptotic cells from the alveoli. In addition to providing anti-inflammatory effects, PAMs secrete a range of cytokines and immunomodulatory factors such as CD200, TGF-β, CD172a, GM-CSF, M-CSF, and IL-10 [8]. Recent studies have shown that innate immune cells, including PAMs, can acquire “trained immunity”—an adaptive-like state mediated by epigenetic reprogramming that enhances responsiveness to subsequent infections [9]. Due to their longevity, vascular proximity, and capacity to retain innate immune memory, PAMs represent a critical model system for studying host antiviral immunity [8,9]. This process involves durable epigenetic modifications, including changes in histone marks, enhancer/promoter accessibility, and regulation by non-coding RNAs, which collectively prime cells for heightened responses upon re-exposure [10]. Although circRNAs have not yet been directly linked to trained immunity or long-term epigenetic memory, related non-coding RNAs have been implicated in this process: van der Heijden et al. reviewed how histone modifications and metabolic rewiring underpin trained immunity in macrophages [10], another study from Flores-Concha et al. highlighted lncRNAs as key regulators of trained immune responses [11], and Nakayama et al. reported that miRNAs such as miR-155 and miR-21 modulate the magnitude of trained immune phenotypes [12]. Given our findings that circRNAs are modulated by PRRSV and interferon stimulation in PAMs, are enriched in immune-related pathways, and form extensive circRNA–miRNA networks, it is plausible that some circRNAs could contribute to epigenetic reprogramming in a trained immunity-like manner [10,11,12].

Circular RNAs (circRNAs) are a diverse and abundant class of endogenous non-coding RNAs (ncRNAs), alongside microRNAs (miRNAs) and long non-coding RNAs (lncRNAs). These molecules originate from non-protein-coding genomic regions and were long considered transcriptional byproducts or “junk” RNA [13]. However, emerging evidence suggests that ncRNAs play essential roles in regulating host–virus interactions. CircRNAs were first observed in RNA viruses via electron microscopy and initially mistaken for viroids [13,14]. With the advent of high-throughput RNA sequencing and advanced computational tools, thousands of circRNAs have now been identified and characterized across different animal species including pigs. Typically derived from protein-coding genes, circRNAs can contain single or multiple exons. Unlike linear RNAs, circRNAs lack 5′ caps and 3′ poly(A) tails, which makes them resistant to RNase R digestion and more stable than linear counterparts [15].

In recent years, circRNAs have been recognized as regulators in competitive endogenous RNA (ceRNA) networks, acting as miRNA sponges and modulating gene expression in diverse physiological and pathological contexts, including infections and cancer [16]. They have also been implicated in antiviral signaling pathways, further supporting their immunological relevance. Some circRNAs contain internal ribosome entry sites (IRESs), which were originally identified in picornavirus mRNAs, enabling cap-independent translation of short peptides or “microproteins” (<100 amino acids) [17]. As circRNAs are produced via back-splicing—a process that competes with canonical linear splicing—they may influence transcript isoform balance and protein output [18]. Owing to their structural stability, resistance to exonucleases, and translational potential, circRNAs have emerged as promising tools for vaccine development. They may serve as both antigen carriers and immunostimulatory adjuvants, enhancing dendritic cell activation and adaptive immune responses [19,20,21].

However, despite growing evidence of the regulatory roles of circRNAs in immunity, the circRNA landscape in porcine alveolar macrophages during PRRSV infection or interferon stimulation remains largely uncharacterized. Previous studies have primarily focused on mRNAs and miRNAs, leaving the contribution of circRNAs to innate antiviral responses unexplored. Moreover, no prior work has systematically investigated the potential of circRNAs to form ceRNA networks or encode bioactive peptides in this context. Filling these gaps is essential to advance our understanding of host immune regulation during PRRSV infection and may inform the development of novel antiviral and vaccine strategies.

The objective of this study was to characterize the genome-wide circRNA responses in porcine alveolar macrophages following stimulation with a PRRS modified live virus (MLV) vaccine and two interferon subtypes (IFN-α1 and IFN-ω5). To achieve this, we performed whole-transcriptome sequencing to profile differentially expressed circRNAs in porcine alveolar macrophages treated with a modified live virus (MLV) vaccine and comparatively, two antiviral interferon subtypes (IFN-α1 and IFN-ω5). To our knowledge, this is the first genome-wide investigation of circRNA expression in response to interferon treatment within the PRRSV context, building upon our prior research on miRNA regulation in the same model [22,23]. Over 1000 differentially expressed circRNAs were identified across treatment groups. Functional annotation using Gene Ontology (GO) and KEGG pathway analyses highlighted potential roles of these circRNA-associated genes in immune regulation and antiviral defense. Further, we evaluated the post-transcriptional regulatory potential and coding capacity of these circRNAs. This included circRNA–miRNA interaction network analysis, prediction of open reading frames (ORFs), N6-methyladenosine (m^6^A) modification sites, and internal ribosome entry sites (IRESs). These analyses aimed to determine whether circRNAs contribute to host antiviral responses through both ceRNA mechanisms and non-canonical translation into bioactive peptides.

## 2. Materials and Methods

### 2.1. Ethics Statement and Animal Cell Sources

No live animal experiments were conducted in this study. Porcine primary cells were obtained from previously cryopreserved samples stored in the laboratory [23]. The Institutional Biosafety and Institutional Animal Care and Use (IBC#1732 and IACUC#5349) committees approved all recombinant DNA procedures and animal procedures. Porcine alveolar macrophages (PAMs) were collected via bronchoalveolar lavage from lungs of six approximately 5-week-old outbred pigs using 300 mL of 10 mM phosphate-buffered saline (PBS, pH 7.4) per animal. Within four hours of collection, PAMs were isolated by centrifugation at 400× *g* for 15 min and further purified via plastic adherence. Freshly isolated or cryopreserved PAMs were immediately stored in liquid nitrogen using Recovery™ Cell Culture Freezing Medium (Invitrogen, Carlsbad, CA, USA) until use. For PRRSV rescue and antiviral assays, the MARC-145 monkey kidney cell line (ATCC) was propagated according to ATCC guidelines or as previously described [23].

### 2.2. Virus Infection and IFN Treatment

PAMs were seeded in 6-well plates in RPMI medium supplemented with 10% fetal bovine serum (FBS, Thermo Fisher Scientific, Waltham, MA, USA) and 1× penicillin-streptomycin (Thermo Fisher Scientific, Waltham, MA, USA). Cells were divided into four treatment groups, each with three independent repeats: (1) mock-stimulated control, (2) modified live virus (MLV) vaccine strain (Ingelvac PRRS MLV, Boehringer Ingelheim Vetmedica, Duluth, GA, USA), (3) porcine IFN-α1, and (4) IFN-ω5 (Kingfisher Biotech, Saint Paul, MN, USA) at a final concentration of 20 ng/mL and incubated for 5 h. PRRSV infection was carried out at a multiplicity of infection (MOI) of 0.1 for 5 h. After infection, cells were washed twice with fresh medium before RNA and protein extraction [24]. For the mock-stimulated group, bovine serum albumin (BSA, ThermoFisher, Waltham, MA, USA) at 20 ng/mL was added to the culture medium as a control. To confirm that the treatments were effective, stimulation of several classical IFN-stimulated genes including IFNARs and IRFs have been analyzed using RT-PCR, as reported previously [23].

### 2.3. RNA Transcriptomic Analysis

Approximately 2.5 × 10^7^ cells pooled from three repeats per treatment were used to extract total RNA using a column-based RNA/DNA/protein purification kit (Norgen Biotek, Thorold, ON, Canada). RNA quality and concentration were assessed with a NanoPhotometer^®^ (IMPLEN, Westlake Village, CA, USA) and Qubit^®^ RNA Assay Kit on a Qubit^®^ 2.0 Fluorometer (Life Technologies, Carlsbad, CA, USA). Only samples with an A260/A280 ratio >1.8 and RNA integrity number (RIN) > 7.0 were selected for library construction. Five micrograms of total RNA per sample underwent rRNA depletion using the Ribo-Zero™ rRNA Removal Kit (Epicentre, Madison, WI, USA), followed by ethanol precipitation. To enrich for circular RNAs, remaining linear RNA was digested using 3 U of RNase R (Epicentre) per microgram of RNA [23].

Library construction was performed using the NEBNext^®^ Ultra™ Directional RNA Library Prep Kit for Illumina^®^. The process includes cDNA synthesis with random hexamers, second-strand synthesis using dUTP substitution, end repair, 3′ adenylation, and adaptor ligation. Fragment sizes (~150–200 bp) were selected using AMPure XP beads, followed by USER enzyme treatment (3 μL, NEB) and PCR amplification. Final libraries were purified with AMPure XP beads and analyzed on an Agilent Bioanalyzer 2100 [23].

As illustrated in Appendix A, raw sequencing reads were pre-processed to remove adapters, poly-N stretches, and low-quality reads, yielding high-quality clean data. Quality metrics including Q20, Q30, and GC content were calculated. The reference genome and annotation files were downloaded, and genome indexing was conducted using HISAT2 (v2.0.4). Paired-end clean reads were aligned to the reference genome using HISAT2. CircRNAs were identified using both find_circ and CIRI2, and visualized using Circos software v0.62-1 [25]. Functional annotation of circRNA host genes was performed via Gene Ontology (GO) enrichment using GOseq [26]. All programs used are listed in Appendix A. Raw sequencing data have been deposited in the NCBI Short Read Archive under BioProject accession number PRJNA882823.

### 2.4. MicroRNA Target Site Prediction and Coding Potential Analysis

Putative miRNA target sites in exonic regions of circRNA loci were predicted using miRanda. To evaluate the coding potential of selected circRNAs, their exon sequences were retrieved from the Ensembl genome database. Internal ribosome entry sites (IRESs) were predicted using DeepIRES, a deep learning-based model trained on canonical mRNA IRES motifs, and DeepCIP, a model optimized for circRNA-specific IRES prediction. Both tools were applied to the full circRNA dataset. N6-methyladenosine modification sites were predicted using deepSRAMP, a high-resolution algorithm based on sequence and secondary structure features for accurate m^6^A site identification [27,28,29].

Control Procedures and Materials: Mock-stimulated porcine alveolar macrophages (PAMs) were used as negative controls to establish baseline circRNA expression profiles. These control cells were cultured under the same conditions as the treatment groups but received no virus or interferon stimulation. All experimental treatments (PRRSV MLV vaccine, IFN-α1, and IFN-ω5) were performed in parallel with their corresponding control samples to ensure consistency. Control samples were processed together with treated samples for RNA extraction, library preparation, and sequencing, thereby minimizing technical variation and enabling reliable differential expression analysis [23].

### 2.5. Statistical Analysis

Statistical analyses were performed using the SAS-v9.4 software package. One-way analysis of variance (ANOVA) followed by Tukey’s post hoc test, as well as a two-sample F-test, were used to evaluate differences between samples/treatments. A *p*-value of <0.05 was considered statistically significant [30].

## 3. Results

### 3.1. CircRNA Compositions and Identification Across the PAM Samples

In the previous study, we explored microRNA differential expression patterns, which provided valuable data within the limited transcriptomic research on the PRRSV-swine mode [23]. Building on our own and others’ prior findings, the present study focuses on investigating differential circRNA responses post antiviral regulation in porcine primary macrophages post stimulation by a PRRS MLV vaccine and two subtypes of porcine interferons (i.e., IFN-α1 and IFN-ω5).

As summarized in Table 1, each group produced over 12 Gb of clean reads, with an error rate consistently around 3%. Phred quality scores above Q20 and Q30—indicating high-confidence base calls—were sufficient to ensure genome-wide coverage and reliable cross-sample comparability for differential expression analyses. Notably, IFN-ω5 treatment generated the largest number of clean reads among all treatments. The length distribution of circRNAs in the four groups exhibit broad but consistent overall pattern, although there are subtle differences among groups. This suggests a shared bias toward the detection of circRNAs in lower length ranges across the datasets.

Then, we aligned the clean reads to the reference genome and annotated circRNAs from each group using a union model to determine the overall classification and composition (Figure 1A). The majority of mapped reads in samples P1, P2, P3, and P4 corresponded to protein-coding transcripts. A considerable fraction of reads (about 20–22%) fell into the “others” category, indicating additional transcript types not covered by the major biotype classifications. Smaller proportions of reads were annotated as long non-coding RNA (1.5–1.9%), ribozyme (1.2–1.4%), and pseudogene (0.1%), with minimal contributions from miscellaneous RNA, mitochondrial rRNA, and scaRNA.

To minimize false positives, we employed two independent tools, CIRI2 and find_circ, which perform two-pass scanning to confirm circRNA signals, including paired chiastic clipping (PCC) and paired-end mapping (PEM). Table 2 presents the top ten gene-abundant novel circRNAs, detailing their genomic positions, strand orientations, lengths (full versus spliced), and associated genes. Where available, known gene names according to standard NCBI nomenclatures (e.g., DMXL2, HUWE1, EML4) are provided.

Following length distribution and circRNA source identification were performed, the results showed that most circRNAs were distributed in the shorter length ranges, with a gradual decrease in abundance as length increased. The trend was consistent across all samples, indicating that the majority of detected circRNAs are relatively short, typically within a few hundred nucleotides (Figure 1B).

As shown in the circular bar plot, the majority of circRNAs in all four groups (P1, P2, P3, P4) were derived from exonic regions, with counts ranging from 4561 to 4973. In contrast, far fewer circRNAs were identified from intronic (124–135) and intergenic (107–125) regions (Figure 1C). It is notable that higher expression of intron-derived circRNAs was detected in P3-IFN-α1 sample comparatively (Figure 1C). Furthermore, circRNAs profiled here were broadly distributed genome-wide across all swine chromosomes with high tractability across the samples (Figure 1D).

### 3.2. General Expression Pattern and Differential Expression of circRNAs

The expression levels of circRNAs across samples were normalized using the TPM (Transcripts Per Million) method [31]. This approach accounts for differences in library size, providing a more accurate comparison of circRNA abundance between samples. A selection of representative circRNAs and their corresponding raw read counts and TPM values across the four experimental groups shown in the Appendix A. Notably, certain circRNAs displayed group-specific expression patterns. For example, novel_circ_0000028 was exclusively detected in the IFN-ω5 group (P4), while novel_circ_0000006 was expressed in all groups but exhibited the highest TPM value in the IFN-α1 group (P3), suggesting condition-specific regulation of circRNA expression. To visualize overall expression patterns, a density distribution plot of log_10_(TPM + 1) values was generated (Figure 2A). The distribution profiles were largely consistent across groups, with minor variations in peak height and width, indicating similar global circRNA expression trends despite treatment-specific differences in individual transcripts. The IFN-ω5 group (P4) showed a slightly sharper density peak, suggesting a tighter expression range. Furthermore, expression levels were categorized into TPM intervals to quantify distribution trends. The vast majority of circRNAs in all groups exhibited high expression levels (TPM ≥ 60), with proportions ranging from 76.49% in P3 to 84.27% in P4. A smaller subset of circRNAs fell within the 0–0.1 TPM interval, accounting for 15.73–23.51% of each group. Notably, no circRNAs were detected in the intermediate expression ranges (0.1–60 TPM), highlighting a bimodal-like distribution where most circRNAs are either highly expressed or minimally expressed across samples. To complement this TPM-based assessment, we also quantified circRNA abundance using back-splice junction reads normalized to total mapped reads (RPM). A box plot of log_10_(RPM + 1) values revealed broadly similar abundance distributions across all four groups (Figure 2B), with most circRNAs detected at low RPM levels and a smaller subset showing higher expression. This RPM-based view confirms that the observed TPM patterns are not driven by differences in sequencing depth, thereby reinforcing the robustness of the circRNA expression profiles across treatments.

In Figure 3, the differential expression and cluster analysis of circRNA differentially expressed (DE-circRNA) in four different samples are shown. The statistic defaults for the differentially expressed circRNA were defined as |log2(Fold Change)| > 1 and q-value < 0.01), and represented by red/blue dots (Figure 3) or bars (Figure 4) for significantly up- or down-regulated circRNAs, respectively, in Volcano plots and heatmap. Volcano plots were generated to visualize the magnitude and significance of expression changes across comparisons (Figure 3). The analysis revealed a substantial number of DE-circRNAs in each condition: 454 up-regulated and 543 down-regulated circRNAs in P2 vs. P1, 457 up-regulated and 570 down-regulated in P3 vs. P1, and 541 up-regulated and 480 down-regulated in P4 vs. P1.

Hierarchical clustering of differentially expressed circRNA patterns across samples is shown in Figure 4. In this heatmap analysis, circRNA expression values were standardized by row using Z-score transformation, and hierarchical clustering was performed with Euclidean distance and complete linkage to group circRNAs with similar expression trajectories. This analysis identified five distinct clusters in the dendrogram, each representing circRNAs with characteristic expression patterns across the treatment groups.

The Venn diagram (Figure 5) illustrates the distribution and overlap of circRNA expression across the four treatment groups (P1, P2, P3, and P4). A core set of 1284 circRNAs was commonly expressed among all groups, suggesting a conserved baseline expression profile. In addition, each group exhibited a distinct subset of uniquely expressed circRNAs, including 47 in P1, 57 in P2, 43 in P3, and 76 in P4.

### 3.3. Gene Ontology and KEGG Analysis of circRNA-Targeted Genes

Gene Ontology (GO) enrichment analysis was performed to investigate the functional roles of circRNA-associated source genes. GO terms—covering biological processes, molecular functions, and cellular components—were mapped using the GO database, and statistically enriched terms were identified using Wallenius’ non-central hypergeometric distribution [26]. The analysis revealed key functional categories significantly associated with circRNA source genes, suggesting their involvement in specific regulatory and antiviral pathways. Figure 6 presents three comparisons: the MLV vaccine (P2), IFN-α1 (P3), and IFN-ω5 (P4) compared to the control group, respectively, which show that top GO terms of significant enrichment (*p*-adj < 0.05).

To investigate the biological functions associated with circRNA source genes, KEGG pathway enrichment analysis was performed across six pairwise treatment comparisons: P4 vs. P1, P4 vs. P3, P4 vs. P2, P2 vs. P1, P3 vs. P2, and P3 vs. P1. Among these, three comparisons (P2 vs. P1, P3 vs. P1, and P4 vs. P1) were selected for focused analysis using the same negative control group (P1). The top 20 significantly enriched pathways for each comparison were visualized in scatter plots (Figure 7), with enrichment levels evaluated by rich factors, q-value, and gene counts.

Across these comparisons, 11 pathways were commonly enriched, indicating shared functional responses among treatments. These included ubiquitin-mediated proteolysis, endocytosis, thyroid hormone signaling, chemokine signaling, Fc gamma R-mediated phagocytosis, and the hepatitis B pathway, among others (Figure 7D).

The two interferon-treated groups, IFN-α1 (P3) and IFN-ω5 (P4), exhibited higher similarity and functional overlap in pathway enrichment compared to the MLV vaccine group (P2). Shared enriched pathways among these groups included leukocyte transendothelial migration, microRNAs in carcinogenesis, Ras signaling pathway, and TNF signaling pathway—all of which are closely associated with the regulation of immune and inflammatory response [32,33]. Notably, IFN-ω5 (P4) displayed slightly higher enrichment levels than IFN-α1 (P3) across these pathways.

### 3.4. CircRNA-miRNA Interactions and Coding Potential

A well-established function of circRNAs is their ability to act as miRNA sponges, modulating the activity of miRNAs and, consequently, the expression of their downstream target mRNAs [15,34]. The effectiveness of this interaction is influenced by the stoichiometric relationship between the number of miRNA binding sites present on the circRNA and those on its target mRNAs [15]. CircRNAs that are highly expressed and harbor multiple binding sites are more likely to engage in competitive endogenous RNA (ceRNA) activity [15,16]. To investigate whether the circRNAs identified in this study may exert such regulatory effects, we utilized the miRanda algorithm to predict potential miRNA binding sites. As illustrated in Figure 8, the circRNA–miRNA interaction network revealed numerous predicted regulatory associations. In the network, circRNAs are represented as blue nodes, miRNAs as yellow nodes, and gray edges denote predicted binding interactions (Figure 8). Due to the complexity of the network, only six representative subsets are displayed for clarity. Notably, ssc-miR-135 is predicted to interact with more than 90 circRNAs, while novel_circ_0003987 shows potential binding with over 20 distinct miRNAs.

CircRNAs possess inherent translational potential, and their ability to encode truncated proteins or micropeptides may contribute to diverse cellular functions [18]. To explore this possibility, we evaluated the protein-coding capacity of a subset of differentially expressed circRNAs identified in this study. A dataset comprising 100 randomly selected circRNAs was subjected to N6-methyladenosine (m^6^A) and internal ribosome entry site (IRES) prediction analyses. Open reading frames (ORFs) were classified based on sequence features: a full ORF was defined as a sequence initiating with a canonical start codon (ATG) and terminating with an in-frame stop codon (TAA, TAG, or TGA), without internal stop codons; a partial ORF included a start codon but lacked a downstream in-frame stop codon. Among the 100 circRNAs analyzed, 51 contained full ORFs, 48 contained partial ORFs, and 1 circRNA lacked any identifiable ORF. For nearly half of the circRNAs with partial ORFs, features such as premature stop codons, delayed start codons, or structural artifacts introduced by back-splicing were frequently observed. These elements may contribute to increased complexity in translational potential by disrupting canonical ORF architecture.

To further assess the potential for cap-independent translation, we employed three computational tools—deepSRAMP, DeepIRES, and DeepCIP—to predict m^6^A modification sites and IRES elements in the same circRNA dataset [27,28,29]. For deepSRAMP, a deep learning-based m^6^A predictor, identified 67 circRNAs with confidence scores ≥ 0.5, and 28 circRNAs with scores ≥ 0.8. Notably, nearly half of these circRNAs harbored multiple m^6^A sites. And from DeepIRES, an IRES prediction tool trained on mRNA features, we predicted 56 circRNAs with scores ≥ 0.5, including 29 with scores ≥ 0.8. We then employed DeepCIP, a circRNA-specific IRES prediction model, identified 32 circRNAs with scores ≥ 0.5, including 6 with scores ≥ 0.8 (Figure 9A).

To explore the potential synergy between m^6^A and IRES-mediated translation, we cross-compared deepSRAMP and DeepIRES predictions. A total of 48 m^6^A-IRES pairs were found to overlap in sequence space. When filtered for confidence scores ≥ 0.5, 7 circRNAs retained both m^6^A and IRES predictions within overlapping regions. Under a stricter threshold (≥0.8), 4 circRNAs remained. These overlaps suggest that m^6^A modifications and IRES elements may act cooperatively to promote cap-independent translation in circRNAs [35].

We subsequently constructed a Venn diagram to illustrate the number of circRNAs predicted with confidence scores ≥ 0.5 by each method (Figure 9B). Among these, 8 circRNAs were identified by all three tools—deepSRAMP, DeepIRES, and DeepCIP—as harboring both m^6^A sites and IRES elements, indicating strong translational potential. Additional overlaps included 29 circRNAs predicted by both deepSRAMP and DeepIRES, 12 by deepSRAMP and DeepCIP, and 5 by DeepIRES and DeepCIP. Each method also yielded unique predictions, with 18 circRNAs identified solely by deepSRAMP, 14 by DeepIRES, and 7 by DeepCIP. We further randomly selected seven circRNA sequences and performed BLAST searches against the NCBI database (Figure 9C). For each circRNA, homologous sequences from 3 to 5 other species were chosen, representing diverse taxonomic groups, including Primates, Rodentia, Carnivora, Pholidota, Artiodactyla, Salmoniformes, and Chiroptera. The resulting pairwise identity heatmap revealed high-identity blocks across these phylogenetically distant species, indicating strong evolutionary conservation. As a representative example, novel_circ_0001883, derived from exons 5–9 of the TASOR gene, was selected for further analysis. According to Ensembl, this circRNA is 392 nucleotides in length and contains several predicted ORFs, one of which encodes a putative 70-amino acid microprotein.

## 4. Discussion

### 4.1. Interpretation of circRNA Composition and Identification in PAMs

CircRNAs are known to participate in various biological processes by acting as miRNA sponges, transcriptional regulators, and even templates for protein translation [21]. Among the four groups, the mock-stimulated group exhibited the lowest proportion of protein-coding reads, whereas the P3-IFN-α1 treatment group showed the highest proportion (76.1%), followed closely by P4-IFN-ω5 (75.3%). Both interferon-treated groups thus had a larger fraction of circRNAs derived from protein-coding genes than the mock control and MLV vaccine groups, suggesting that IFNs may stimulate the expression of circRNAs derived from protein-coding genes.

From the compositional profiles, these results demonstrate the predominance of protein-coding sequences in the samples. The circRNA-seq data provided a comprehensive and reliable classification and composition of lung immune cells, laying a strong foundation for downstream investigations. We used two independent tools, CIRI2 and find_circ, and this dual approach ensured accurate detection of circRNA boundaries, correct length assignment, and high-confidence reference-based annotation. Collectively, these data highlight the diversity of circRNAs detected and underscore their back-splicing evidence and associations with different host genes across multiple experimental treatments [18,36].

According to current studies, circRNAs originate from different genomic regions, including exonic, intronic, and intergenic sequences [15,37]. As shown in Figure 1C,D, our findings confirm that exon-derived circRNAs are the dominant form across treatments, while intron-derived circRNAs are differentially regulated, indicating a divergence within a mainly conserved biogenesis preference and potential regulatory significance in host immune responses [15].

### 4.2. Biological Implications of circRNA Expression Patterns and Differential Profiles

Certain circRNAs displayed clear group-specific expression patterns, such as novel_circ_0000028 (exclusive to IFN-ω5) and novel_circ_0000006 (highly expressed under IFN-α1), suggesting condition-specific regulation of circRNA expression. The density distribution profiles were largely consistent across groups, with only minor variations in peak shape. This indicates similar global circRNA expression trends despite treatment-specific differences in individual transcripts, and the sharper peak in IFN-ω5 suggests a tighter expression range. The overall bimodal-like expression pattern, where most circRNAs are either highly expressed or minimally expressed, highlights strong transcriptional control of circRNA levels in PAMs.

Volcano plot analysis revealed that each IFN subtype and MLV vaccine triggered distinct circRNA expression profiles, with IFN-ω5 (P4) inducing the largest number of upregulated circRNAs, and IFN-α1 (P3) associated with the highest number of downregulated transcripts. This suggests that different antiviral treatments modulate circRNA production through distinct regulatory pathways.

In the hierarchical clustering analysis (Figure 4), five discrete expression clusters were identified. Notably, the first cluster includes six circRNAs exhibiting markedly lower expression levels in PAMs treated with the MLV vaccine (P2) and atypical IFN-ω5 (P4), compared to those treated with the conventional antiviral IFN-α1 (P3) and the control group (P1). This observation highlights that even within interferon treatments, different IFN subtypes can exert diverse regulatory effects, functioning either as suppressors or stimulators of circRNA expressions in PAMs [38]. The second cluster clearly demonstrates a similar circRNA expression pattern among PAMs treated with antiviral IFN-α1 (P3), IFN-ω5 (P4), and the MLV vaccine (P2). All three treated groups exhibit significantly elevated circRNA expression compared to the control (P1), suggesting analogous antiviral regulatory mechanisms between these IFN subtypes and the vaccine strain [23,38]. These findings further underscore the potential importance of circRNA-mediated responses in IFN- and MLV-driven antiviral pathways, particularly at the immune cell level [39]. The third cluster comprises circRNAs implicated in the regulation of balanced antiviral responses and immunometabolic processes, which are notably active during T-cell activation and periods of increased energy demand in immune cells. Within this cluster, PAMs treated with antiviral IFN-α1 (P3) displayed significantly higher circRNA expression compared to the other three groups, which showed comparatively mild or suppressed expression levels.

The fourth and fifth clusters consistently demonstrate that PAMs-treated with IFN-ω5 (P4) exhibit substantial modulation of circRNAs implicated in key immune response path-ways. More than ten representative circRNAs were significantly upregulated in the P4-treated group. These circRNAs are derived from genes such as JAK2, AKIRIN2, SMARCA5, and TRPM7, which are involved in diverse immunological processes including immune cell activation, transcriptional regulation, NF-κB coactivation in innate im-munity, mRNA stabilization, and exosomal targeting. The marked changes in circRNA expression observed under IFN-ω5 stimulation underscore its robust antiviral potential. These findings are consistent with recent in vivo studies reporting enhanced antibody responses, decreased viral titers, and favorable cytokine profiles in IFN-ω5-treated animals [40]. These findings collectively emphasize the crucial role of early immune activation mediated by IFN-ω5 in effectively combating PRRSV infection as reported previously [40,41].

From the Venn diagram, these unique and overlapping circRNA populations highlight the complexity of circRNA-mediated regulation and suggest that specific subsets may play important roles in differential immune responses to various treatments [18,42]. Comparative expression analyses further revealed that both IFN-α1 and IFN-ω5 treatments induced a greater number of differentially expressed circRNAs relative to the control group, with IFN-ω5 eliciting the most pronounced effect. Specifically, IFN-ω5 treatment led to a substantially higher number of upregulated circRNAs compared to both the MLV vaccine and IFN-α1 treatments. This distinct expression profile mirrors previously reported patterns observed in miRNA expression studies, suggesting a coordinated and subtype-specific regulatory response of non-coding RNAs, particularly circRNAs, and may associate with IFN-ω5-mediated superior antiviral activity [40].

### 4.3. Functional Significance of circRNA-Targeted Genes in GO and KEGG Pathways

The GO enrichment results suggest that circRNAs play key regulatory roles in cellular homeostasis and immune-related metabolic activities. In all three comparisons, the most significantly enriched biological processes (BP) included cellular metabolic processes, protein modification processes, and macromolecule biosynthetic processes, suggesting that circRNAs play key regulatory roles in cellular homeostasis and immune-related metabolic activities. Within the cellular component (CC) category, enriched terms such as nucleus, intracellular organelle, and membrane-bound organelle were consistently identified, indicating that circRNAs likely exert their regulatory functions within critical intracellular compartments [18,43]. In the molecular function (MF) category, RNA binding, nucleotide binding, and catalytic activity terms were highly enriched, supporting the hypothesis that circRNAs may participate in RNA regulatory networks and immune signaling modulation [44].

Between groups, MLV vaccine (P2) treatment showed strong enrichment in nucleic acid binding and cellular biosynthetic processes, indicating broad transcriptional and post-transcriptional regulation upon the vaccine MLV stimulation. IFN-α1 (P3) treatment prominently enriched organic cyclic compound binding and transcription regulatory activities, reflecting substantial transcriptional reprogramming induced by IFN-α1. In contrast, IFN-ω5 (P4) treatment exhibited a distinct enrichment for methylation-related processes, including RNA and protein methylation, suggesting a unique fine-tuning mechanism of innate immune activation mediated by IFN-ω5 [40]. Notably, IFN-ω5 (P4) treatment exhibited a relatively higher proportion of genes associated with metabolic and biosynthetic processes compared to MLV (P2) and IFN-α1 (P3), implying a more profound cellular activation status under the unconventional IFN-ω5 stimulation. Additionally, unique to IFN-ω5 treatment (P4), there was a notable enrichment in processes related to RNA and protein methylation, such as methyltransferase activity, histone methylation, and RNA methylation. These modifications are known to be critical for fine-tuning innate immune activation, antiviral gene expression, and epigenetic regulation [32,33,44].

The KEGG enrichment results further support these findings. Many enriched pathways are involved in immune system processes, especially antiviral and antitumor responses, as well as signal transduction mechanisms regulating transcription, apoptosis, and cell cycle progression [20,45,46,47]. Notably, IFN-ω5 group consistently displayed slightly higher enrichment levels than IFN-α1 across immune-related pathways, suggesting stronger immune modulation. P3-treated PAMs uniquely activated TGF-beta signaling and chronic myeloid leukemia pathways, indicating possible links to immunosuppressive or leukemogenesis-related processes, while P4-treated PAMs showed broader enrichment in pathways such as NF-κB signaling pathway, B cell receptor signaling pathway, propanoate metabolism, and choline metabolism in defensive immunity.

Mounting evidence from diverse porcine viral infections supports our findings, with particularly strong parallels in lung macrophages. Chen et al. showed that PEDV infection of intestinal epithelial cells induced widespread circRNA remodeling and predicted circRNA–miRNA regulatory networks, while Li et al. reported extensive circRNA reprogramming in PK-15 cells during Pseudorabies virus infection and implicated antiviral miRNAs such as miR-210 and miR-340 as circRNA targets [36,48]. Similarly, another study (Porcine circovirus type 2 infection activates NF-κB pathway and cellular inflammatory responses through circPDCD4/miR-21/PDCD4 axis in porcine kidney 15 cell) also demonstrated that circPDCD4 promotes NF-κB activation and inflammation during PCV2 infection through a circPDCD4–miR-21–PDCD4 axis, providing direct mechanistic evidence of circRNA–miRNA regulation in viral immunity [49]. Most notably, Dai et al. revealed a functional ceRNA network in porcine alveolar macrophages infected with swine influenza virus, where novel_circ_0004733 and TCONS_00166432 sponge miR-10391 to regulate MAN2A1, thereby modulating key innate immune sensors (RIG-I, TLR7, NLRP3), cytokine secretion, and viral gene expression [50]. This work directly demonstrates that circRNAs can actively shape antiviral responses in the same cell type used in our study, reinforcing our observation that PRRSV and interferon stimulation extensively remodel circRNA expression and engage immune-related circRNA–miRNA networks in PAMs.

These findings suggest that IFN-ω5 not only contributes to innate and adaptive immune responses, apoptosis, and inflammation, but may also influence metabolic-immune interactions, oncogenic signaling, and immune cell behavior [40,45]. This diversity underscores the multifaceted regulatory role of IFN-ω5 in antiviral defense and immune modulation at both molecular and cellular levels [19,23,40,45,47].

### 4.4. Regulatory Roles and Translational Potential of circRNAs via miRNA Interactions

The extensive interaction patterns observed in the circRNA–miRNA network highlight the complex post-transcriptional regulatory roles circRNAs may play, particularly in antiviral responses. Similar mechanisms have been reported in several RNA viruses, including hepatitis C virus (HCV), porcine epidemic diarrhea virus (PEDV), avian leukosis virus (ALV), and human immunodeficiency virus (HIV), further supporting the relevance of circRNA–miRNA networks in viral pathogenesis and immune modulation [36,51,52,53]. The ORF analysis revealed that a large proportion of circRNAs harbor full or partial coding potential, suggesting that back-splicing may preserve coding sequences capable of translation. The presence of premature stop codons or disrupted ORF architecture in some circRNAs may add regulatory complexity and modulate translational outcomes. The m^6^A and IRES prediction results provide further evidence of the translational versatility of circRNAs, as the overlap of m^6^A and IRESs within the same circRNAs suggests that these features may act cooperatively to promote cap-independent translation through non-canonical mechanisms involving epitranscriptomic regulation and structured RNA elements [35,44]. The conservation analysis strengthens this interpretation: high pairwise identity of selected circRNAs across diverse vertebrate species suggests they may retain translational potential and perform conserved biological functions [15,54].

Collectively, these findings emphasize that certain circRNAs may encode bioactive peptides or proteins. For instance, novel_circ_0001883 contains a predicted ORF encoding a 70-amino acid microprotein. If translated, such peptides could either compete with full-length protein isoforms or exert distinct functional roles. Although these observations point to intriguing regulatory potential, they remain putative. Our predictions of circRNA–miRNA interactions and translational activity are currently based on computational analyses and have not yet been experimentally validated. To substantiate these findings, future studies will employ targeted experimental approaches such as luciferase reporter assays to verify circRNA–miRNA interactions, RNA immunoprecipitation (RIP)-seq to detect associated RNA–protein complexes, and ribosome profiling or mass spectrometry to assess the active translation of circRNA-derived peptides. Such functional validations will be essential to confirm the mechanistic relevance of the circRNAs identified in this study and to advance our understanding of their roles in PRRSV infection and interferon-mediated antiviral responses.

## 5. Conclusions

This study provides a comprehensive transcriptomic analysis of circRNA expression in porcine alveolar macrophages (PAMs) following treatment with a PRRS modified live virus (MLV) vaccine and two interferon subtypes (IFN-α1 and IFN-ω5). By profiling circRNA expression under these distinct antiviral conditions, we identified both conserved and treatment-specific expression patterns, suggesting that circRNAs play diverse regulatory roles in antiviral immune responses. Bioinformatics analyses, including GO and KEGG pathway enrichment, revealed that circRNA-associated genes participate in key immune-related processes such as cytokine signaling, apoptosis, inflammation, and immune cell activation [18,37,44,46]. Additionally, we explored the translational potential of selected circRNAs through open reading frame (ORF) prediction, m^6^A methylation analysis, and identification of internal ribosome entry sites (IRESs). The co-localization of m^6^A motifs and IRES elements within specific circRNAs, such as novel_circ_0000141, supports the possibility of cap-independent translation, potentially giving rise to microproteins with functional relevance. These circRNA-derived microproteins may contribute to antiviral defense either through novel mechanisms or by competing with full-length protein isoforms [33,55].

Collectively, our findings capture the early phase of the circRNA response in porcine macrophages upon the IFN and virus treatments, which reinforce the regulatory significance of circRNAs in IFN-mediated antiviral responses and highlight IFN-ω5 as a particularly potent stimulator of circRNA expression and immune activation [23]. Furthermore, our results highlight the differential expression patterns elicited by IFN subtypes, paving the way for identifying the most effective antiviral candidates in subsequent studies. These findings also align with recent in vivo studies, which demonstrated that IFN-ω5 plays a critical role in the early phase of immune activation and cytokine-mediated antiviral responses, supporting its promise as a target for immunomodulatory and therapeutic strategies [40]. Moreover, circRNAs showing strong immune enrichment or regulatory activity may be explored as potential adjuvant targets or delivery vehicles in next-generation PRRSV vaccines, complementing antigen-based approaches and improving the induction of durable immunity. Overall, this study establishes that circRNAs are integral regulators of early antiviral responses in porcine macrophages, providing new molecular insights and potential targets to improve PRRSV control [6,20].

## Figures and Tables

**Figure 1 viruses-17-01307-f001:**
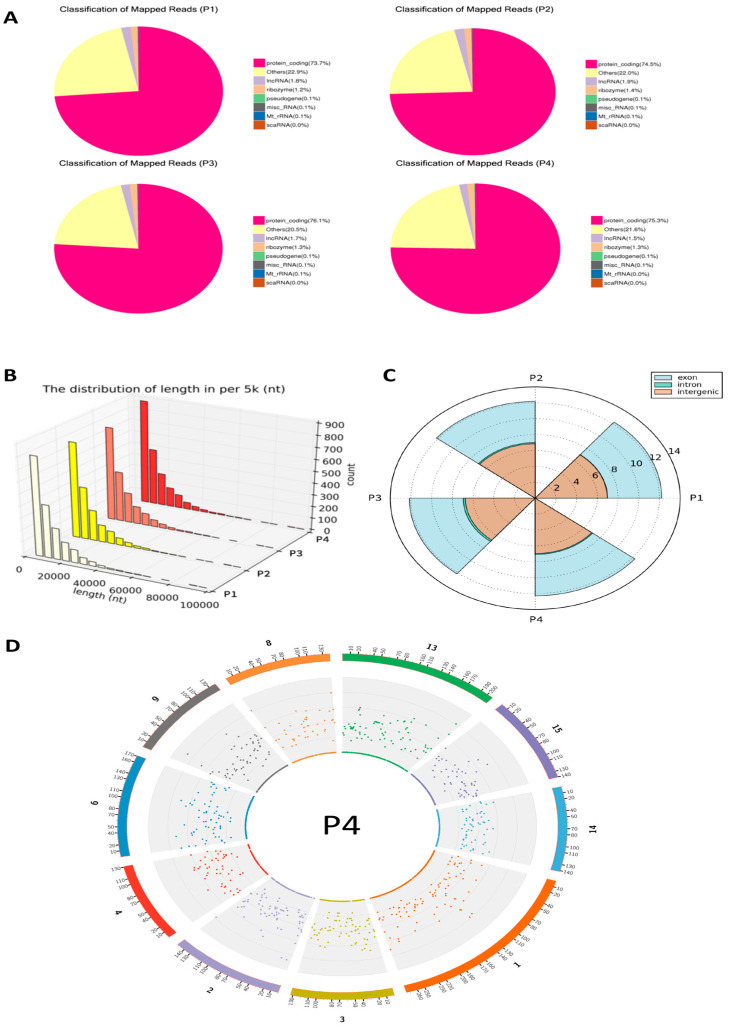
Distribution and classification of mapped reads of annotated circRNAs based on transcript types, length, genetic regions and chromosome location. (**A**) Genome distribution of mapped circRNA reads. Categories are distinguished by color. Across all groups, approximately 75% of mapped reads were derived from protein-coding transcripts. (**B**) The length distribution of circRNAs is generally comparable across the samples, the colors transition from light to dark to represent different samples. (**C**) Genetic sources of circRNAs, derived primarily from back-spliced exons or intergenic regions, and to a much lesser extent from intronic regions. (**D**) The density of total mapped reads in each chromosome was calculated, and 10 chromosomes or scaffolds were selected for cross-sample comparison (shown for P4, others see in Appendix A).

**Figure 2 viruses-17-01307-f002:**
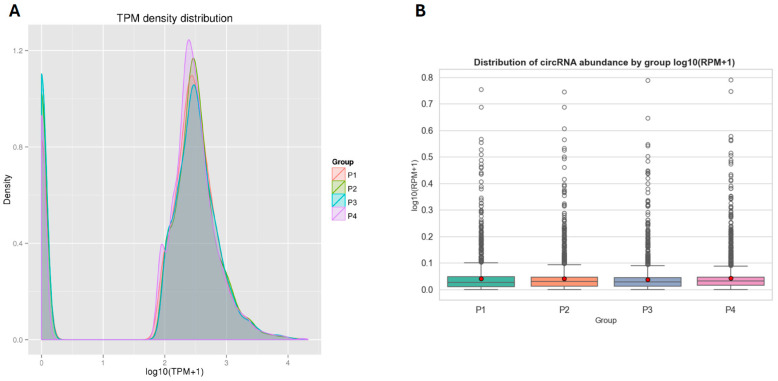
(**A**) Density distribution plot of circRNA expression levels (log_10_(TPM + 1)) across the four experimental groups (P1, P2, P3, and P4). All groups exhibited a bimodal-like distribution, with the majority of circRNAs either highly expressed (TPM ≥ 60) or minimally expressed (TPM ≤ 0.1). (**B**) Distribution of circRNA abundance (log_10_(RPM + 1)) across groups (P1–P4), based on back-splice junction reads normalized to total mapped reads per sample. The expression profiles are highly similar among groups, displaying minor variations in peak heights and widths.

**Figure 3 viruses-17-01307-f003:**
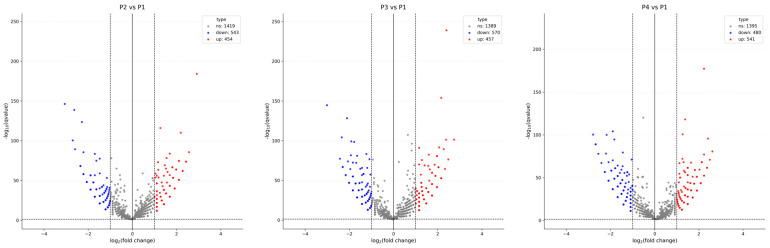
Volcano plots depicting differential circRNA expression across pairwise comparisons of four experimental samples (P1, P2, P3, and P4). Differentially expressed circRNAs were identified based on thresholds of |log_2_(Fold Change)| > 1 and q-value < 0.01. Significantly upregulated circRNAs are shown in red, downregulated circRNAs in blue, and non-significant circRNAs in gray. Each plot illustrates clear distinctions in circRNA expression patterns among the compared groups. P4 has the most up-regulated transcripts and P3 has the highest number of down-regulated transcripts.

**Figure 4 viruses-17-01307-f004:**
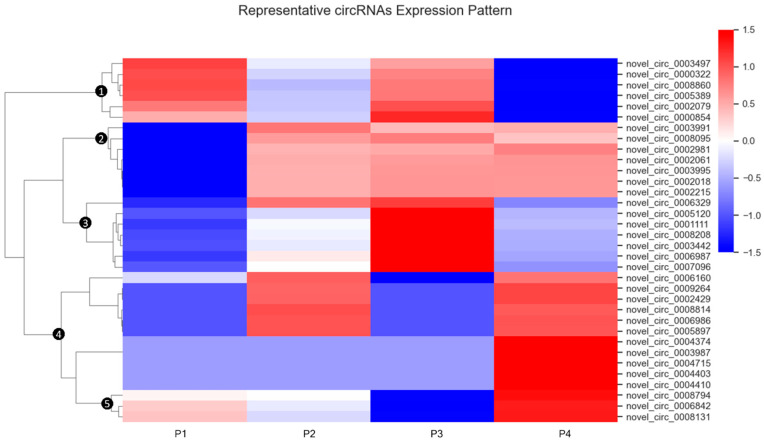
Expression values were standardized by row using Z-score transformation to enable comparison across circRNAs. Hierarchical clustering was performed using Euclidean distance and complete linkage to group circRNAs with similar expression trajectories. CircRNAs were grouped into five distinct clusters (①–⑤), labeled on the dendrogram. Expression intensities are visualized with a color gradient, where red represents higher relative expression and blue represents lower relative expression. This analysis highlights distinct circRNA expression dynamics associated with each treatment group. The complete dataset of identified circRNAs is provided in Appendix A.

**Figure 5 viruses-17-01307-f005:**
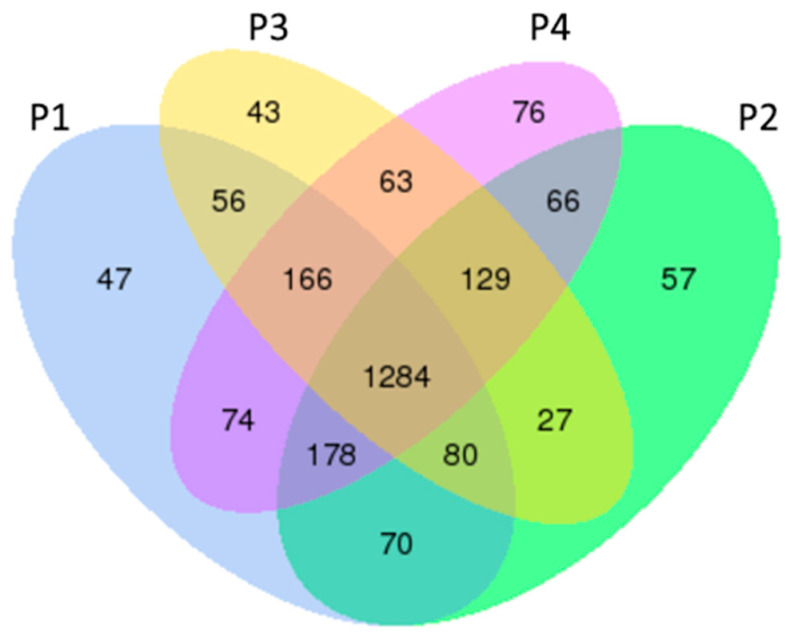
Venn diagram showing the distribution and overlap of circRNA expression among four treatment groups (P1, P2, P3, and P4). A core set of 1284 circRNAs was expressed in all groups, while each group also exhibited uniquely expressed circRNAs, indicating both conserved and condition-specific circRNA expression patterns.

**Figure 6 viruses-17-01307-f006:**
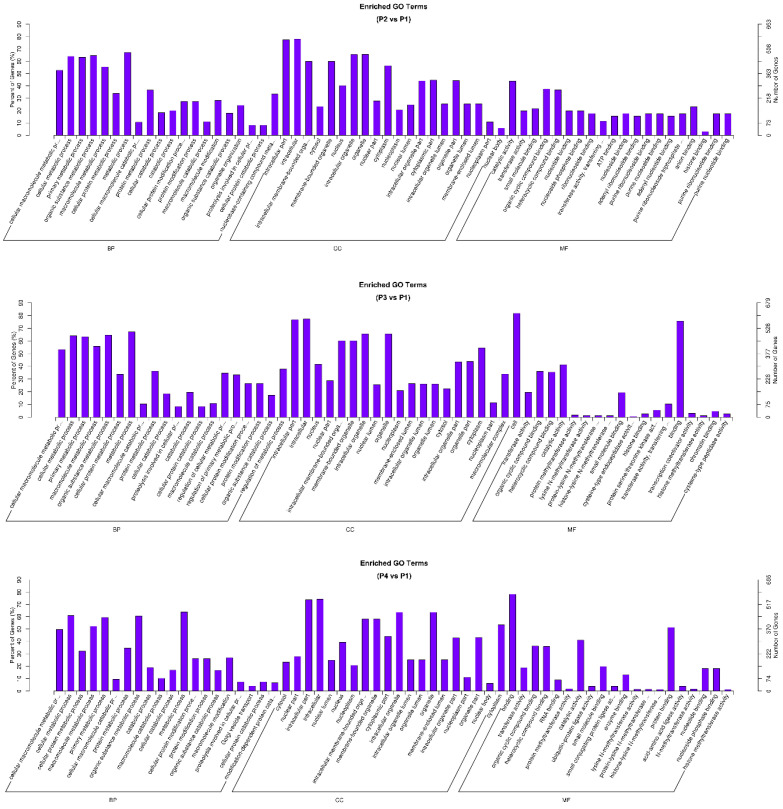
Gene Ontology (GO) enrichment analysis illustrating significantly enriched functional terms (adjusted *p*-value < 0.05) associated with circRNA source genes identified in pairwise comparisons of treatment groups—MLV vaccine (P2), IFN-α1 (P3), and IFN-ω5 (P4)—relative to the control group (P1). The analysis categorizes enriched GO terms into biological processes (BP), cellular components (CC), and molecular functions (MF), based on Wallenius’ non-central hypergeometric distribution. Distinct enriched GO terms in each comparison highlight circRNA-associated genes’ involvement in specific biological pathways and functions, suggesting their potential roles in immune regulation and antiviral responses. Some long terms are truncated in the axis labels, but their full names are provided here for clarity: cellular macromolecule metabolic process; intracellular membrane-bounded organelle; transferase activity, transferring phosphorus-containing groups; purine ribonucleoside triphosphate binding; protein-lysine N-methyltransferase activity; histone-lysine N-methyltransferase activity; modification-dependent protein catabolic process; small conjugating protein ligase activity.

**Figure 7 viruses-17-01307-f007:**
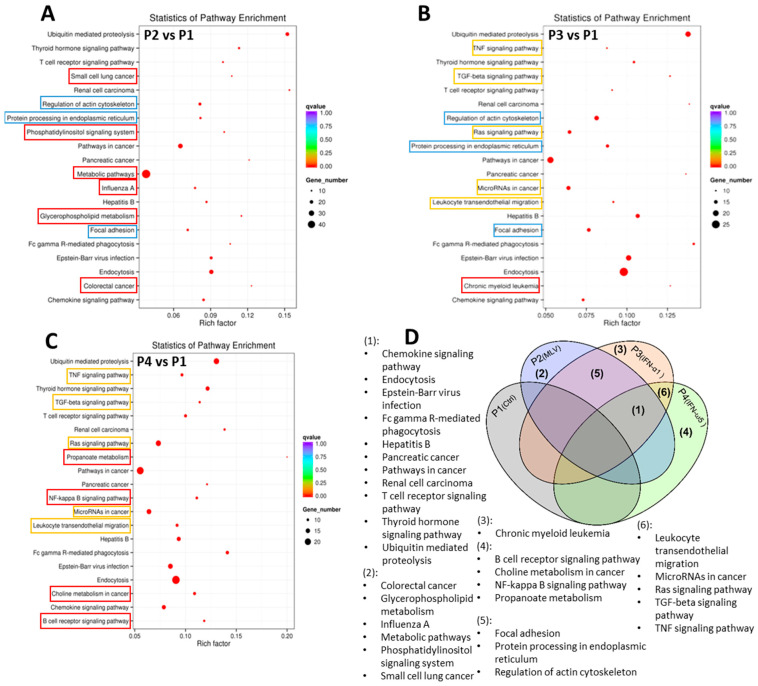
Scatter plots depicting KEGG pathway enrichment analysis of circRNA-associated source genes from six pairwise treatment comparisons (P4 vs. P1, P4 vs. P3, P4 vs. P2, P2 vs. P1, P3 vs. P2, and P3 vs. P1). (**A**–**C**) Each plot illustrates the top 20 significantly enriched pathways. The *X*-axis indicates the enrichment factor, while the *Y*-axis lists enriched KEGG pathways. Dot sizes correspond to the number of genes associated with each pathway, and colors reflect the significance level (q-value). Pathway enrichment analysis highlights specific biological processes and signaling pathways potentially regulated by circRNAs under distinct treatment conditions, suggesting their involvement in immune responses and antiviral mechanisms. Pathways framed indicated unique pathways enriched in each compared pair only (Red frame) or some compared pairs (Yellow or Blue frame). (**D**) Venn diagram to summarize pathways: (1) common pathways unframed in (**A**–**C**), or (2) to (6) unique pathways unframed in (**A**–**C**) enriched in the different comparisons, respectively, as indicated.

**Figure 8 viruses-17-01307-f008:**
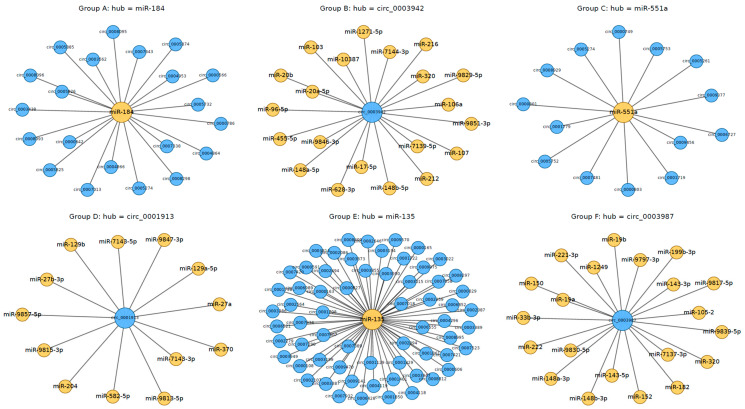
circRNA–miRNA interaction network illustrating predicted regulatory associations between circRNAs (blue nodes) and miRNAs (yellow nodes). Gray edges represent predicted binding interactions derived from miRanda analysis. Due to network complexity, only a representative subset of interactions is visualized. Subnetworks from six groups are presented as zoomed-in examples, with the corresponding circRNA–miRNA connections listed in Appendix A. And the full network is provided in the Appendix A.

**Figure 9 viruses-17-01307-f009:**
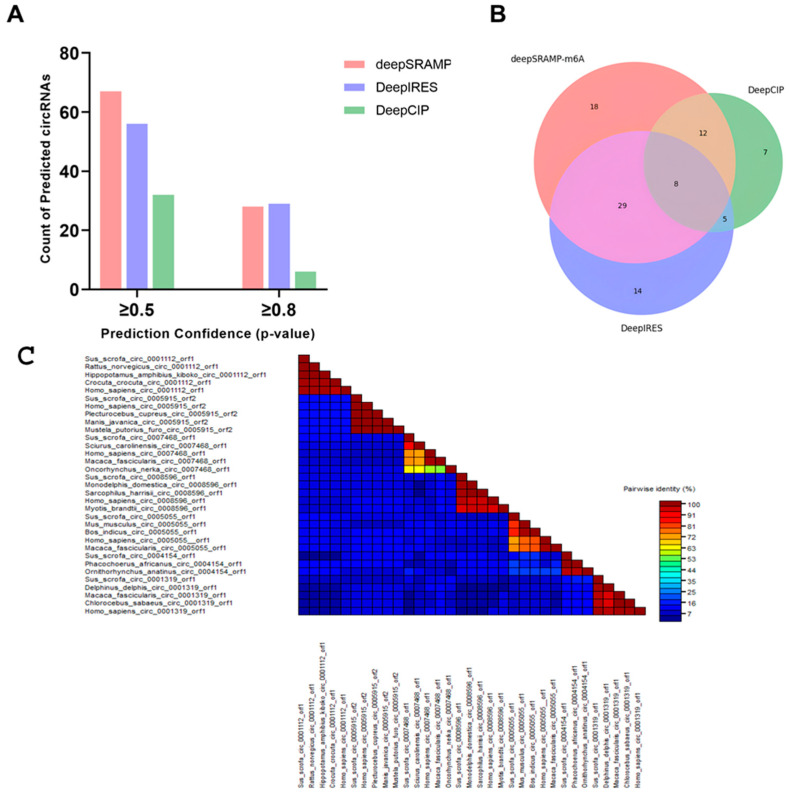
(**A**) Bar plot showing the number of unique circRNAs predicted by each tool (deepSRAMP, DeepIRES, DeepCIP) at two confidence thresholds (≥0.5 and ≥0.8). (**B**) Venn diagram illustrating the overlap of high-confidence circRNAs (confidence ≥ 0.5) identified by the three prediction models. Each circle represents circRNAs predicted by one method, and overlaps indicate shared predictions, suggesting functional convergence. (**C**) Pairwise identity (%) plots among amino acid sequences of small ORFs predicted from representative porcine (Sus scrofa) circRNAs and their conserved homologs from other animal species indicating cross-species conservation of translational potential of these circRNAs in biological regulation. Comparison and plot drawing were performed using a SDT program.

**Table 1 viruses-17-01307-t001:** Summary of sequencing quality metrics across four experimental groups *.

Sample	Raw Reads	Clean Reads	Clean Bases	Error Rate	Q20	Q30	GC Content
**P1 (Ctrl)**	90284642	88711244	13.3 G	0.03	97.56	93.53	53.74
**P2 (MLV)**	83084248	81698196	12.3 G	0.03	97.28	92.88	53.74
**P3 (IFN-α1)**	85481978	83905840	12.6 G	0.03	97.67	93.8	53.34
**P4 (IFN-ω5)**	91860194	90253536	13.5 G	0.03	97.46	93.27	52.2

* Each group (P1–P4) yielded over 12 Gb of clean sequencing data with consistently low error rates (0.03%). Phred quality scores above Q20 and Q30 indicated high sequencing accuracy, ensuring reliable downstream analysis. GC content ranged from 52.2% to 53.74%, showing minimal variation across samples.

**Table 2 viruses-17-01307-t002:** Summary of the Top Ten circRNAs Identified in the Dataset.

CircRNA_id	Chr.	Start Position	End Position	Strand	Full Length	Spliced Length	Gene Id	Gene Name
**novel_circ_0001167**	12	60348041	60353600	-	5559	367	ENSSSCG00000018049	*SLC5A10*
**novel_circ_0003958**	1	120176021	120185745	+	9724	480	ENSSSCG00000032517	*DMXL2*
**novel_circ_0005029**	2	151040091	151052939	+	12,848	370	ENSSSCG00000014440	*HMGXB3*
**novel_circ_0009109**	9	19852497	19861801	-	9304	560	ENSSSCG00000014913	*PICALM*
**novel_circ_0000601**	11	533608	534240	-	632	303	ENSSSCG00000009569	*PSPC1*
**novel_circ_0000083**	10	24760508	24779098	-	18,590	284	ENSSSCG00000010928	*KDM5B*
**novel_circ_0001614**	13	200007333	200012649	+	5316	206	ENSSSCG00000012055	*MORC3*
**novel_circ_0004019**	1	121731740	121738042	+	6302	452	ENSSSCG00000004646	*ATP8B4*
**novel_circ_0007230**	6	109117106	109141734	-	24,628	411	ENSSSCG00000003712	*OSBPL1A*
**novel_circ_0009337**	9	92047368	92049234	+	1866	297	ENSSSCG00000032241	*GPNMB*

## Data Availability

The data presented in this study are openly available in NCBI Short Read Archive under BioProject accession number PRJNA882823 [NCBI Short Read Archive] [https://www.ncbi.nlm.nih.gov/sra/?term=PRJNA882823] [PRJNA882823] (accessed on 23 August 2025).

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
