# Peer review of "Comparative Transcriptomics Reveals Novel and Differential Circular RNA Responses Underlying Interferon-Mediated Antiviral Regulation in Porcine Alveolar Macrophages"

_viruses, 2025, doi:10.3390/v17101307_

Round 1
Reviewer 1 Report
Comments and Suggestions for Authors
Dear editor, the manuscript prepared by Dr. Jiuyi Li et al., is an excellent and timely study that provides a significant contribution to the fields of veterinary virology, immunology, and non-coding RNA biology. The research is well-designed, the methodologies are robust and state-of-the-art, and the findings are novel and compelling. The manuscript is clearly written and logically structured. The identification of IFN-ω5 as a potent inducer of a unique circRNA response is a particularly strong and impactful finding.
This is the first genome-wide analysis of circRNA responses to interferon treatment in the context of PRRSV and porcine alveolar macrophages. This fills a critical knowledge gap and opens new avenues for understanding antiviral immunity.
My comments are:
The authors need to review the paper for typos and errors, also table 5 is mentioned in the text of the manuscript, with the title, but the table itself is missing. The authors jump from table 1 to 5. Please correct all these issues.
Please provide a high resolution figures, especialy regarding figure 1, it's almost unreadable.
Page 2, Line 58-60: The sentence "Recent studies have shown that innate immune cells... can acquire 'trained immunity'" is very relevant. It might be worth briefly speculating in the discussion whether the observed circRNA changes could contribute to this epigenetic reprogramming.
Page 17, Funding: "Please add:" appears to be a placeholder that should be removed before publication.
One more thing to strengthen your paper, this is a common point for purely sequencing-based studies. The miRNA interactions and translational potential are compellingly predicted but not experimentally validated. While this is understood for a foundational transcriptomics paper, mentioning the need for future functional validation (e.g., luciferase reporter assays, RIP-seq, ribosome profiling, mass spectrometry) in the discussion would strengthen the manuscript and better set reader expectations.
Author Response
Please find Point-by-Point Responses in the file oploaded below

Reviewer 2 Report
Comments and Suggestions for Authors
The manuscript addresses a timely and interesting topic, namely the circRNA response in porcine alveolar macrophages (PAMs) to PRRS-MLV and interferons (IFN-α1, IFN-ω5). The overall design is promising, and the integration of functional annotation and translational potential adds clear value. I offer the following comments that the authors may wish to consider to further improve the manuscript.
I encourage the authors to specify the number of replicates per condition and clarify whether samples came from different animals. Indicating n per group, the origin of the samples (animal, age, sex when applicable), and whether batches were included would provide readers with more confidence in the reported changes and in how variability was managed.
It would strengthen the manuscript if circRNA abundance were not expressed solely as TPM, especially given the RNase R treatment. The authors could consider quantifying reads spanning the back-splice junction for each circRNA and comparing across samples using a simple normalization such as reads per million.
Some conclusions are currently phrased in a causal way, for example that IFN-ω5 tunes immunity via methylation or that certain circRNAs have translation. The authors could soften this language by using expressions such as “our data suggest,” “enrichments consistent with,” or “translational potential.”
It would also be valuable to confirm that the treatments were effective. For interferons, showing activation of one or two interferon-responsive genes by qPCR would suffice. For MLV at 5 h, a minimal signal of entry or infection, or a solid reference in the same system, would provide useful context. If these validations cannot be performed, the authors could acknowledge this as a limitation in the Discussion so that circRNA changes are interpreted cautiously.
Since 5 h is an early time point for circRNA accumulation, the authors might wish to note this explicitly as a limitation and clarify that the study captures the early phase of the response.
In the GO and KEGG analyses, I encourage the authors to de-emphasize broad cancer-related pathways that may feel out of context in this system and to focus instead on the main results related to immunity, signaling and trafficking. Oncology-related pathways can still be mentioned but kept secondary.
The circRNA–miRNA network appears very broad and dense. Highlighting 5–10 circRNA–miRNA pairs with clear changes and adequate abundance would make the results easier to interpret, while additional interactions could be provided in the Supplementary Material.
The results on translational potential (ORF, IRES, m6A) are striking but remain predictive. Framing them as candidates for future validation would align the claims with the type of evidence presented.
The authors are encouraged to standardize terminology by using PAMs consistently (rather than alternating with AMs) and preferring “circRNAs derived from protein-coding genes” instead of “protein-coding circRNAs.”
For abundance distributions, the authors could complement the density plot with a box or violin plot by group using the proposed metric (back-splice junction reads per million).
For the volcano plot, adopting a conventional color scheme (for example, red for upregulated, blue for downregulated and gray for non-significant points) would improve readability and match field standards.
In the heatmap, a brief note indicating the clustering method and how the scale was standardized would make the analysis clearer and more reproducible.
Author Response

(The authors gave the same response as above.)

Reviewer 3 Report
Comments and Suggestions for Authors
The authors have reported a transcriptomics analysis of macrophages of pigs after infection with PRRS virus. There is significant novelty in this study, and the authors are to be commented for this excellent conceptualization.
However, the manuscript requires extensive improvement before acceptance. The relevant points are marked herebelow.
- Please explain clearly the gaps in the literature that would be filled with the publication of this work.
- Please define the objectives clearly and concisely.
- M & M. Please add details of manufacturers for all equipment and consumables used in this study.
- M & M. Please describe clearly the control procedures and the control materials that you used in this study. Preferably this should be added in a new subsection separately.
- Whilst the existing figures are excellent indeed, I feel that there is still room for reducing the text in the body of the manuscript and increasing the number of tables and figures presenting results. This will help readers to grasp better the findings of this study.
- I suggest that the Discussion is separated from the Results section.
- Also, I suggest that the Discussion is divided into sub-sections to improve the flow of reading.
- Please add a short paragraph discussing the analogies of this work with other similar studies performed with other viral pathogens of pigs.
- How the results can influence the understanding of porcine immunology and the development of vaccine production against the disease?
- The references are OK, although the authors could have added some recent references from China, which are relevant to this field.
- Please reduce speculation in the final section and also please close the text with a strong take-home sentence.
Overall. Revision and re-evaluation for final acceptance.
Author Response

(The authors gave the same response as above.)

Round 2
Reviewer 2 Report
Comments and Suggestions for Authors
I would like to thank the authors for paying close attention to the reviewers’ comments.
I only have one additional remark regarding the figures, they appear noticeably pixelated. I am not certain whether this is due to a conversion error, but I suggest that the authors provide them in a minimum resolution of 300 DPI to ensure optimal visualization, particularly when readers zoom in. After this remark, I consider that the authors have adequately addressed all the suggested comments, improving both the narrative clarity and the understanding of the data in their manuscript.
Therefore, I believe the manuscript can now be considered for publication.
Author Response
Thank you for the constructive comments on our manuscript, "Viruses Manuscript ID viruses-3862862."
In response to the feedback, we have taken the following actions:
- All figures have been replaced with high-resolution versions within the main manuscript file.
- The high-resolution figures are also provided as separate PDF files for clarity.
- As suggested, the manuscript has undergone extensive proofreading, including review by two native English speakers, to enhance language quality and clarity.
Reviewer 3 Report
Comments and Suggestions for Authors
No further comments, acceptance recommended.
Author Response

(The authors gave the same response as above.)
